# Cryptic Lineage and Genetic Structure of *Acanthopagrus pacificus* Populations in a Natural World Heritage Site Revealed by Population Genetic Analysis

**Md Rakeb-Ul Islam [1,2,*], Katsunori Tachihara [1] and Hideyuki Imai [1]**

[1] Laboratory of Marine Biology and Coral Reef Studies, Faculty of Science, University of the Ryukyus, Nishihara 903-0213, Okinawa, Japan
[2] Department of Fisheries and Marine Science, Noakhali Science and Technology University, Noakhali 3814, Bangladesh
* Correspondence: mrakeb_nstu@yahoo.com; Tel.: +88-01318023706

**Abstract:** Recent studies have revealed extensive genetic differentiation among some populations of marine taxa that were previously believed to be essentially homogeneous because larvae are widely dispersed in ocean currents. *Acanthopagrus pacificus* is a commercially and ecologically important teleost fish that is endemic to shallow coastal waters and estuaries of some tropical and sub-tropical areas in the West Pacific Ocean. Here, we examined genetic structure and the inferred demographic history of *A. pacificus* populations from mtDNA control region sequence data. A 677–678 base-pair fragment was sequenced from 159 individuals sampled at three localities across the West Pacific Ocean. Haplotype diversity was high, ranging from 0.915 to 0.989, while nucleotide diversity was low to medium, ranging from 0.8% to 2.60%. Analysis of molecular variance (AMOVA) showed significant genetic subdivision ($F_{ST} = 0.155$, $p < 0.05$) among sampled populations while pairwise $F_{ST}$ estimates also revealed strong genetic differentiation among populations indicating that gene flow was restricted. Two distinct cryptic lineages were identified that were estimated to have diverged during the Pleistocene. In summary, contemporary factors including regional oceanic currents and self-recruitment are considered to have played significant roles in producing the population structure in this fish. In particular, the genetic information generated in the current study will allow appropriate fisheries management and conservation strategies to be developed for this important local fish in the waters around Iriomotejima Island, a World Heritage site.

**Keywords:** demographic history; control region; Pleistocene; genetic structure; cryptic lineage





## 1. Introduction

Knowledge about levels and patterns of genetic diversity and population structure of target fish species can provide important information for formulating appropriate resource management and conservation strategies [1,2]. In general, it is presumed that the majority of marine fish taxa will display only limited or no population structure due to their large population sizes, high fecundity, and few barriers to natural gene flow in the marine environment, even over very large geographical scales [3–5]. In addition, many marine fish produce relatively long-lived pelagic larvae capable of passive dispersal over long distances via ocean currents. Dispersal of larvae can have a significant influence on the population dynamics, stability, and resilience of wild populations and species expansion, as well as their potential to be connected over wide geographical scales [6–8]. Modern applications of molecular genetics-based approaches to stock structure analysis of marine taxa, however, have modified traditional views by detecting extensive population genetic structure in some marine fish both at macro-geographic and micro-geographic scales, even at a range of ten to a few hundred kilometres [5,9–11]. Population genetic differentiation

may arise due to isolation of populations over a long time period due to natural geographical (e.g., continents) and/or hydrological barriers (e.g., oceanographic fronts, divergence of water masses, salinity gradients or temperature clines) in the marine habitat [12]. In addition, significant levels of local self-recruitment have also been reported in some marine taxa despite having potential for planktonic larval dispersal [13,14]. If populations are self-replaced due to local larval retention over a substantial time frame, then genetically differentiated sub-populations can develop in marine organisms even at moderate spatial scales [15–17]. Some marine biologists have also predicted the occurrence of high levels of self-recruitment of marine larvae due to the effect of diffusion; differential mortality rates and lack of suitable adult habitat may function as strong selection pressures for larvae to remain in the natal habitat [18,19]. Ocean currents may also act as fronts that form strong barriers to wide dispersal of larvae and adults leading to genetic differentiation among populations [20–24]. Recent seascape genetic studies have suggested that limited dispersal of larvae may also occur due to on-shelf water circulation factors, as in some regions with a wide continental shelf, and the majority of the larvae never reach the shelf-edge boundary due to currents [25]. Genetically subdivided self-recruiting populations potentially can have great impacts on evolutionary and ecological patterns [26]. So in general, a genetically homogeneous populationis expected among both geographically adjacent and distant populations if the larvae of a species are widely dispersed via marine water. In parallel, significant genetic structure among populations is expected if the larvae of a species are retained in the natal habitat because natural processes result in high rates of self-recruitment due to behavioural or physical oceanographic mechanisms.

Sparid fishes in the genus *Acanthopagrus* are commercially important fishes that occur widely in shallow coastal waters and estuaries across the Indo-West Pacific Ocean [27]. The target sparid in the present study (*Acanthopagrus pacificus*) has a natural distribution that incorporates the western Pacific including the Ryukyu Archipelago in Japan, Taiwan Island, southern China, Vietnam, the Philippines, Thailand, Malaysia, Indonesia, Papua New Guinea and Australia [28]. *A. pacificus* also utilizes brackish waters in estuaries and inlets, coastal rivers and tidal creeks around Iriomotejima Island in southern Japan [29]. Recently Iriomotejima Island was registered as a natural world heritage site by UNESCO due to a high biodiversity value and a very high percentage of endemic species despite theisland's small size (289 km$^2$) [30]. Since *A. pacificus* is an important and dominant member of the local fish assemblage around Iriomotejima Island, the understanding of the genetics of this sparid fish will allow appropriate management and conservation strategies to be developed. Very little is known about the genetic diversity and genetic structure of this species despite both the ecological and fisheries importance of this species.

Hence, the aim of the current study was to investigate genetic diversity levels and patterns of population structure in *A. pacificus* across the West Pacific using a mitochondrial DNA control region sequence analysis. This information can assist in management, conservation and sustainable utilization of this species as well as other endemic fish species in the same region.

## 2. Materials and Methods

### 2.1. Sample Collection and DNA Extraction

In order to assess genetic diversity levels and population structure, a total of 159 *A. pacificus* specimens were collected from the IriomotejimaIsland (Nakamagawa River, INP, n = 53; Funaura Bay, AFP, n = 61), and Aparri fish market, Luzon Island, the Philippines (PPA, n = 45) (Figure 1, Table 1). Samples were collected from Iriomotejima Island according to Okinawa Prefecture Board of Education permission No. 82. Total genomic DNA was extracted from approximately 50 mg of fresh, frozen, or ethanol-preserved fin tissue using the proteinase K, phenol-chloroform procedures with TNES-8M urea buffer [31].

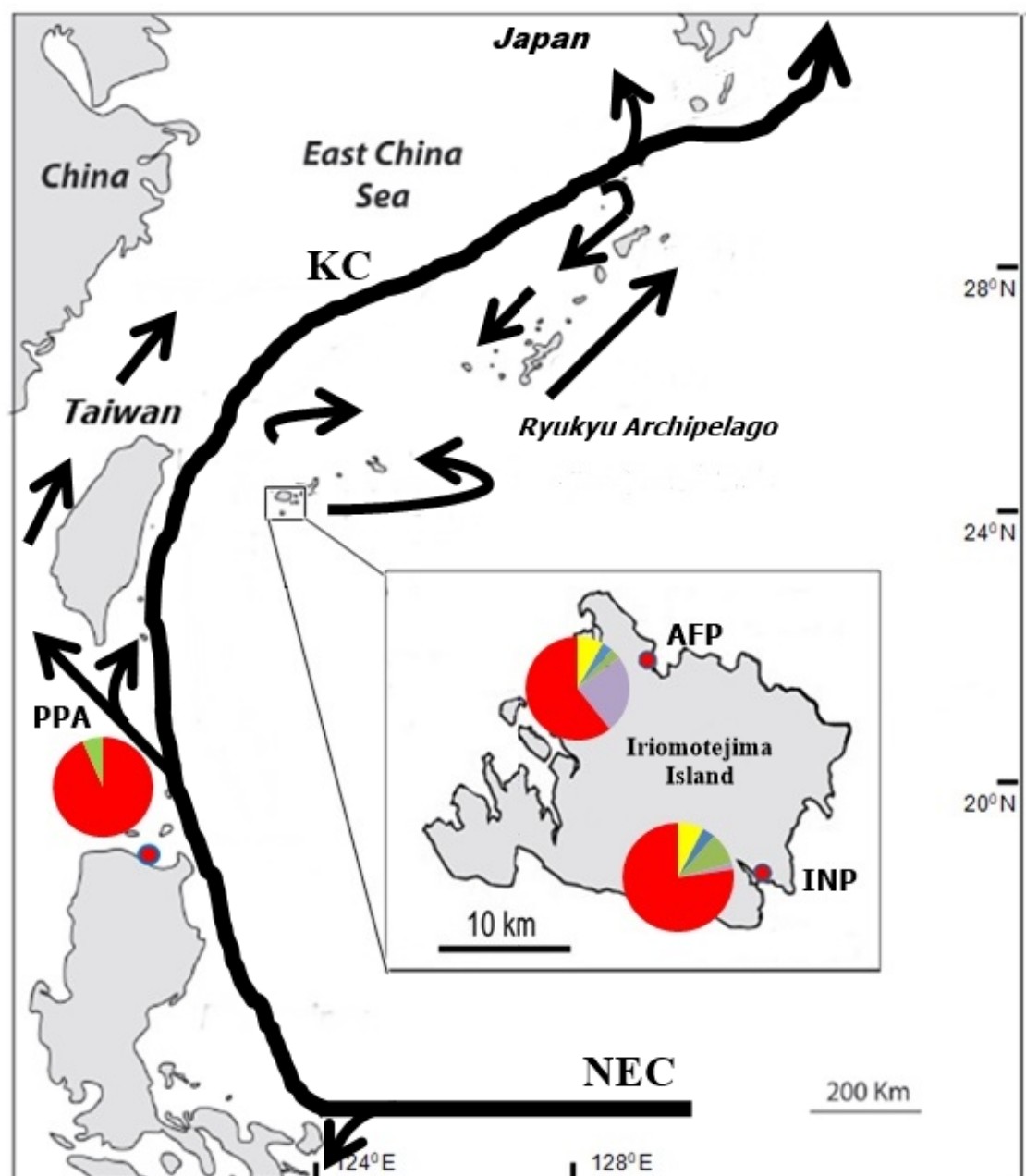

**Figure 1.** Sampling locations are denoted as small red circle and haplotype frequencies are represented in pie graph. KC, Kuroshio Current; NEC, North Equatorial Current; AFP, Funaura, Iriomotejima Island; INP, Nakamagawa, Iriomotejima Island; PPA, Aparri, Luzon Island, Philippines.

**Table 1.** Summary of sampling sites, sample size (N), and genetic diversity of *A. pacificus*.

| Localities | Code | Geographic Coordinate | N | Year | $N_h$ | *h* (SD) | *k* (SD) | $\pi$ (SD) in % |
|---|---|---|---|---|---|---|---|---|
| Funaura Bay, Iriomotejima Island, Japan | AFP | 24°24′17.6″ N 123°48′40.3″ E | 61 | 2020 | 21 | 0.915 (0.229) | 5.608 (2.729) | 0.80(0.40) |
| Nakamagawa River, Iriomotejima Island, Japan | INP | 24°17′31.6″ N 123°52′00.6″ E | 53 | 2020 | 20 | 0.942(0.014) | 11.022 (5.089) | 1.60(0.80) |
| Aparri fish market, Luzon Island, Philippines | PPA | 18°35′08.26″ N 121°63′78.62″ E | 45 | 2012 | 36 | 0.989 (0.007) | 17.720 (8.017) | 2.60(1.30) |

Note: $N_h$, Number of haplotypes; *h*, haplotype diversity; *k*, mean pairwise nucleotide difference; SD, Standard deviation; $\pi$, nucleotide diversity.

### 2.2. Amplification and Sequencing

The mtDNA control region was amplified via polymerase chain reaction (PCR) using the primers Pro-goby (5′-CCACCBCTRRCTCCCAAAGC-3′) [32], 12SarH (5′-ATAGTGGGG TATCTAATCCCAGTT-3′) [33]. All reactions were conducted in a 25 μL total volume and the following reagents were added to each PCR microtube: 0.5 μL of template DNA, 25 pmolof each primer, 12.5 μL of reaction buffer and 0.5 μL of Gflex Taq DNA polymerase (Takara Bio, Shiga, Japan). Each sample was adjusted to 25 μL with distilled $H_2O$. The following basic thermocycler settings for PCR were performed: initial plate heating at 94 °C (2 min) and 30 cycles of denaturation at 94 °C (1 min), annealing at 58–61 °C 30 s, and extension at 72 °C (1.5 min), and final extension at 72 °C (10 min) using the thermal cycler GeneAmp 9700 (Applied Biosystem, Carlsbad, CA, USA). PCR products were checked via electrophoresis on 1% gel agarose and then purified using a pre-sequencing kit (USB Co., Cleveland, OH, USA). Amplified DNA was sequenced on an ABI 3700 genetic analyser (Applied Biosystems, Carlsbad, CA, USA) using the Big Dye Terminator Cycle Sequencing kit ver. 3.1 (Applied Biosystems, Carlsbad, CA, USA). All unique haplotype sequences were deposited in the DNA Data Bank of Japan (DDBJ accession number LC643484-LC643555).

### 2.3. Genetic Diversity Analysis

After sequencing, sequence data were edited and aligned initially using ClustalW [34] applying default parameters in MEGA version 7.0 [35], following which alignments were optimised manually. The number of haplotype ($n_h$), haplotype diversity (*h*) [36], number of private haplotype and nucleotide diversity (*π*) [37] were calculated for each sampled populations using DnaSP 6.0 software (Barcelona, Spain) [38]. Number of polymorphic sites (*K*) for each population was also calculated using Arlequin version 3.5 [39].

### 2.4. Population Genetic Structure

Arlequin version 3.5 [39] was then used to estimate pairwise $F_{ST}$ values between population pairs and to estimate the extent of genetic differentiation among populations. Level of significance was corrected via the Bonferroni adjustment. Distribution of variance within and among populations was determined via analysis of molecular variance (AMOVA) in Arlequin version 3.5 [39].To assess relationships among haplotypes with the number of individuals, a minimum spanning network (MSN) based on the number of nucleotide substitutions among haplotypes was constructed using Popart (Population Analysis with Reticulate Trees) version 1.7 [40] and the network was prepared manually. Neighbour-joining (NJ) tree analysis was then performed in MEGA version 7 program [35]. Genetic divergence between lineages were also calculated using Kimura's two-parameter (K2P) model [41] in MEGA version 7 program [35]. Phylogenetic tree of control region haplotypes was constructed using neighbour-joining (NJ) algorithms of *A. pacificus* where a closely related species, *A. schlegelii* [42] was used as outgroup.

### 2.5. Pattern of Historical Demography

Neutrality tests and mismatch distribution analysis were performed to infer possible population expansion events and to test deviations from neutral model of evolution. Tajima's *D* [43] and Fu's *Fs* [44] indexes were estimated in Arlequin version 3.5 [39]. Significance of estimates were tested via random permutation with 1000 replicates. The demographic expansion parameter (*τ*) and population size before ($\theta_0$) and after expansion ($\theta_1$) were then estimated for each sampled population. A goodness of fit test was conducted to test the validity of a sudden expansion model, using a parametric bootstrap approach based on the sum of the square deviation (SSD) between the observed and expected mismatch distributions. Harpending's raggedness index (HRI) [45] was then estimated for each population. Small raggedness values are typical of an expanding population, while high values are observed among stationary or bottlenecked populations [45,46].

Bayesian skyline analyses, implemented in BEAST v. 1.7.4 [47] were then performed to estimate changes in effective population size over time for the entire sample and for each locality. This method can indicate past demographic changes via a comparison with current patterns of genetic diversity within a sampled population [48]. To check for convergence, we executed multiple independent runs for 30,000,000 iterations under an HKY nucleotide substitution model applying a strict molecular clock. A mutation rate of $3.6 \times 10^{-8}$ per site per year (3.6%/Myr) reported for the mtDNA control region of a teleost [49] and an average generation time of four years [50] were used in this analysis. Effective population sizes were checked and confirmed as >200 for each parameter in order to avoid autocorrelation of parameter sampling [51]. Skyline plots were then generated in Tracer v.1.5 (Edinburgh, UK) [52].

## 3. Results

### 3.1. DNA Variation and Genetic Variability

A 677–678 bp fragment was sequenced from 159 samples of *A. pacificus* collected from three localities. In total 72 unique haplotypes were identified by comparing all individuals' base sequences; 68 haplotypes were restricted to their sampling locality. 37.21% of all individuals possessed three common haplotypes (Supplementary information, Table S1). Haplotype diversity indices (*h*) for each population ranged from 0.915 (AFP) to 0.989 (PPA) and nucleotide diversity (π) in each population varied from 0.8% (AFP) to 2.6% (PPA) (Table 1).

### 3.2. Inferred Population Demographic History

Tajima's *D* and Fu's *Fs* were used to determine if a selecting sweep or balancing selection had impacted sampled populations in the past. Results of two statistical tests for the AFP and INP populations did not deviate significantly from neutral molecular evolution (Table 2). Fu's *Fs* test, however, was significant for the PPA population while Tajima's *D* did not depart significantly from a neutral evolution model. Mismatch distributions were generated for each sampled population to evaluate if their distributions conformed to a model of sudden expansion [51]. Mismatch distributions were unimodal for the Iriomotejima populations (AFP and INP) and bimodal for the Philippine population (PPA) (Figure 2). A Bayesian skyline plot revealed a rapid increase in effective size dating to around 30,000 years before present (ybp) for the Philippine population at the end of the Pleistocene. In contrast, both Iriomotejima populations (AFP and INP) showed plot slopes that were not significantly different from zero based on BEAST analysis (Figure 3). However, past demographic expansion of *A. pacificus* populations was observed in pooled samples for all sampled populations based on Bayesian skyline analyses.

**Table 2.** Results of the mismatch analysis and neutrality tests for *A. pacificus* populations.

| Locality | Demographic Parameters | | | | Test of Goodness of Fit | | Neutrality Test | |
|---|---|---|---|---|---|---|---|---|
| | τ | Obs. | $\theta_0$ | $\theta_1$ | SSD (*p*) | Ragged (*p*) | Tajima's D (*p*) | Fu's Fs (*p*) |
| Funaura | 3.728 | 5.609 | 0.000 | 14.087 | 0.006 (0.250) | 0.017 (0.740) | −1.346 (0.063) | −3.805 (0.132) |
| Nakamagawa | 0.000 | 11.022 | 10.800 | 6887.570 | 0.0237 (0.980) | 0.007 (1.000) | 0.647 (0.790) | 0.134 (0.590) |
| Philippines | 8.303 | 17.720 | 15.089 | 174.061 | 0.0054 (0.460) | 0.004 (0.590) | −0.119 (0.512) | **−11.169** (0.004) |

τ, expansion parameter; obs. mean, mismatch observed mean; $\theta_0$ mutation parameter before ($\theta_0$) and after ($\theta_1$) expansion; Ragged, raggedness index of Harpending; Significant values (*p* < 0.05) are shown in bold. Corresponding *p*-values for each parameter are in parentheses.

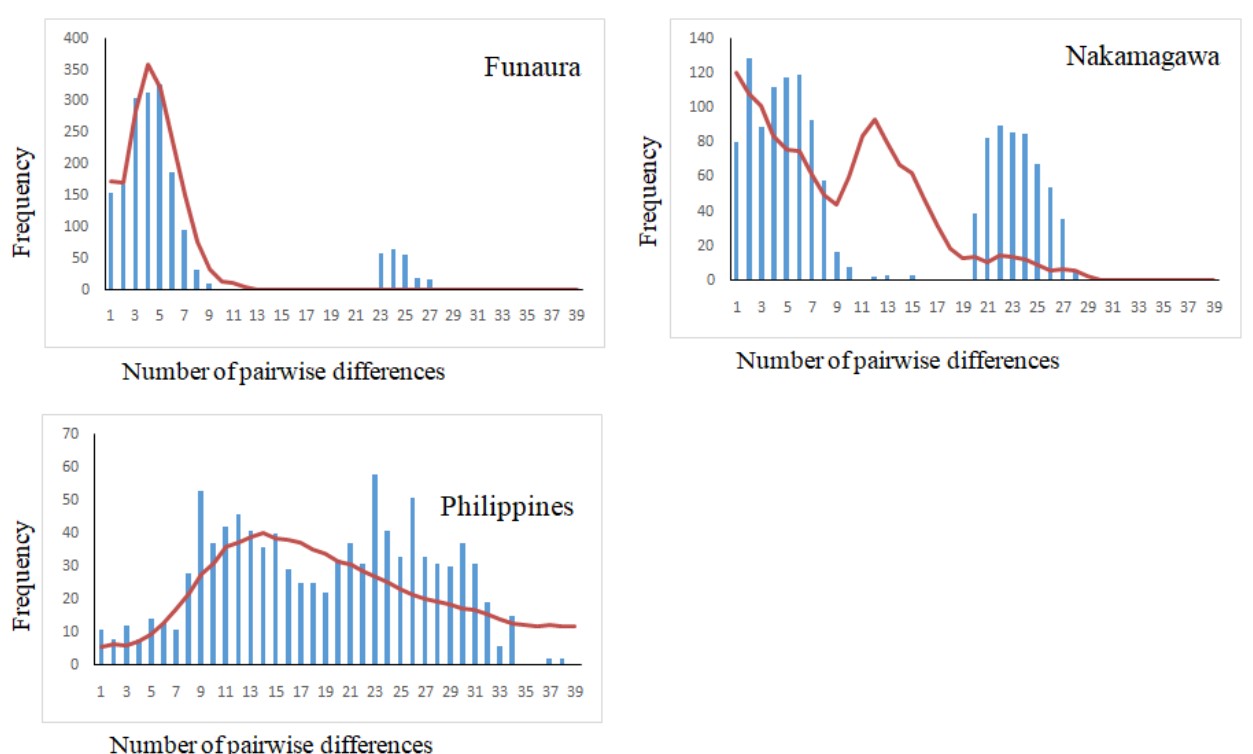

**Figure 2.** Pairwise mismatch distribution for *A. pacificus* by location. The observed frequencies (bar) and the expected mismatch distributions under a model of sudden expansion (solid line) were shown for the mitochondrial DNA control region.

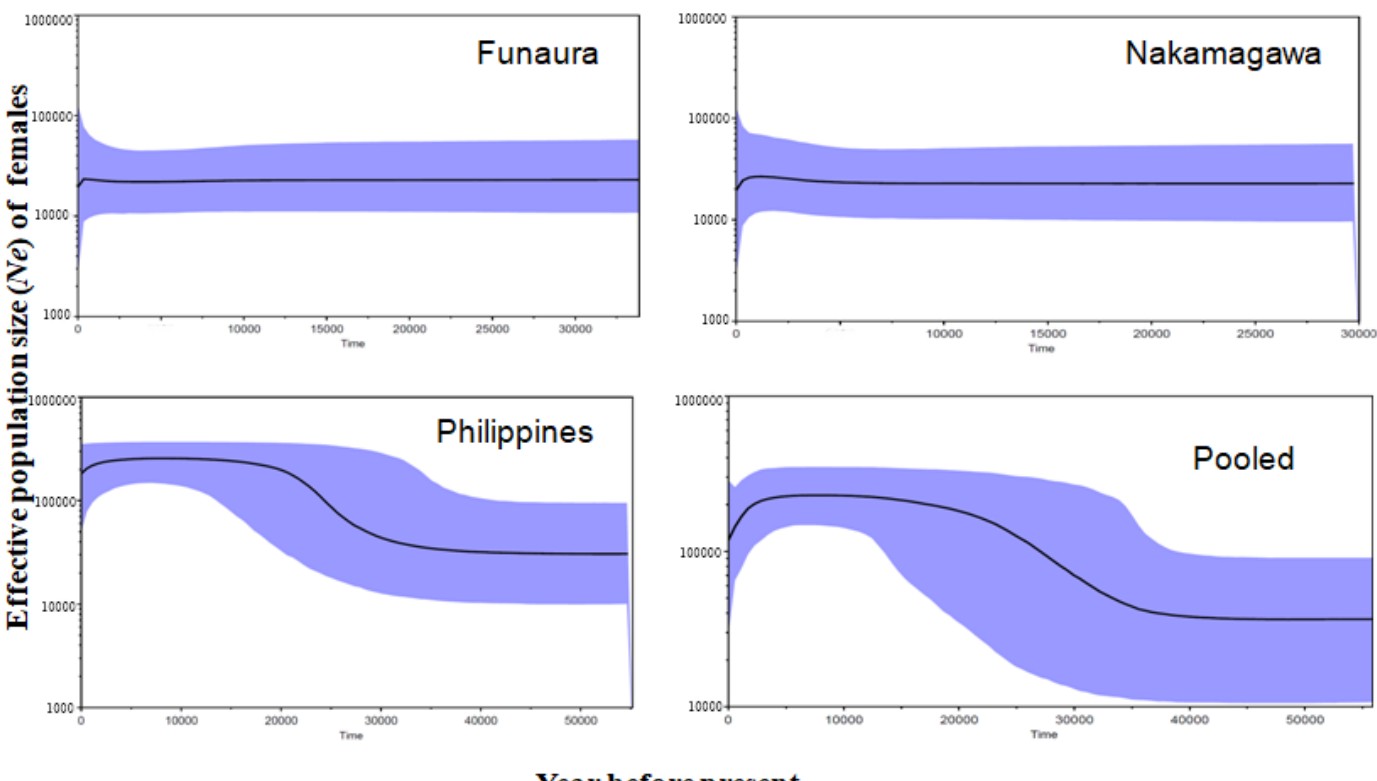

**Figure 3.** Bayesian skyline plots analysis for *A. pacificus* populations implemented in BEAST. The thick solid line depicts the median estimate and margins of the shape area represent the highest 95% posterior density intervals.

### 3.3. Population Genetic Structure

Pairwise $F_{ST}$ values estimated between population pairs are presented in Table 3. Significant pairwise $F_{ST}$ values between populations were observed in all population combinations ($p < 0.05$) with the largest pairwise $F_{ST}$ values observed between Funaura (AFP) and the Philippine (PPA) populations. Strong genetic structure was also evident in the AMOVA analysis (Table 4). AMOVA results showed significant genetic differentiation among all populations ($F_{ST} = 0.155$; $p < 0.05$). Significant variation was also found within populations amounting to 84.54% ($p < 0.05$). Two sub-networks were evident based on minimum spanning tree (MST) haplotype network analysis, with 3.43% genetic divergence observed between these two sub-networks (Figure 4). Two lineages (Lineage A and Lineage B) were also revealed based on NJ analysis (Figure 5).

**Table 3.** Pairwise $F_{ST}$ values (below diagonal) for the mitochondrial DNA control region among three locations of *A. pacificus*. *p*-values ($p < 0.05$) are shown (above diagonal) after Bonferroni correction.

| Locality | AFP | INP | PPA |
|---|---|---|---|
| AFP |  | 0.000 | 0.000 |
| INP | 0.135 |  | 0.000 |
| PPA | 0.237 | 0.089 |  |

**Table 4.** Summary of analysis of molecular variance (AMOVA) for mitochondrial DNA control region sequence of *A. pacificus*. Significance values ($p < 0.05$) are shown in bold. *df*, degrees of freedom; % var., percentage of variance.

| Comparisons | Source of Variance | *df* | % var. | Fixation Indices |
|---|---|---|---|---|
| All populations | Among populations | 2 | 15.46 | **0.155** |
| | Within populations | 156 | 84.54 | |

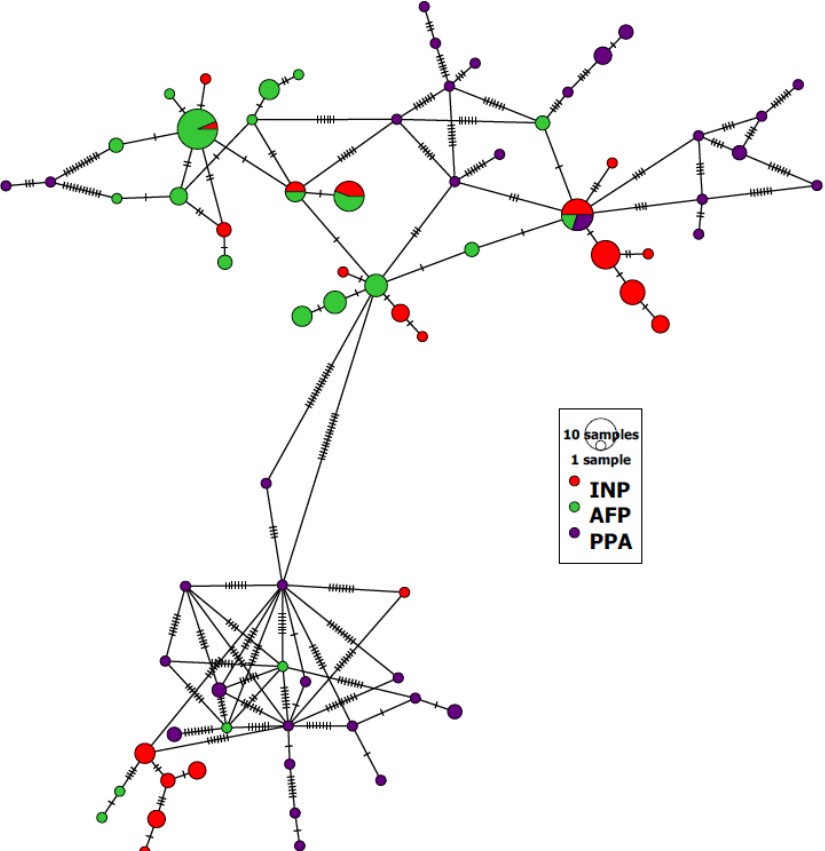

**Figure 4.** Minimum spanning tree representing the haplotype network of all *A. pacificus* samples. Coloured circles represent haplotypes (with circle size scaled by frequency). Hash marks between haplotypes represent one substitution step (i.e., one nucleotide difference).

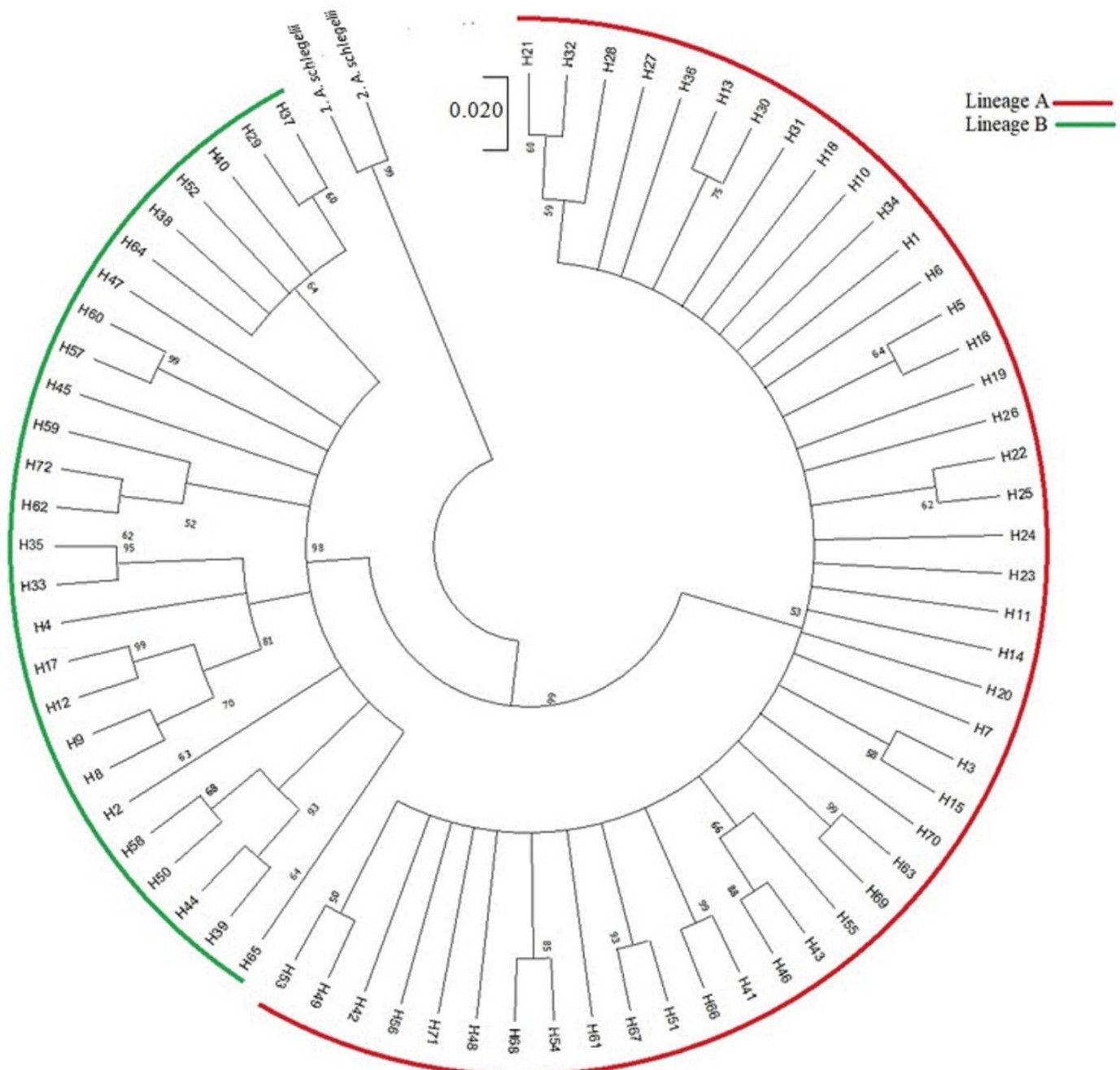

**Figure 5.** Phylogenetic tree of control region haplotypes constructed using neighbour-joining (NJ) algorithms of *A. pacificus* with *A. schlegelii* as outgroup.Haplotypes (H) information of all three localities isgiven in Table S1.

## 4. Discussion

### 4.1. Genetic Diversity

The analysis of genetic diversity in the sampled *A. pacificus* populations showed high haplotype diversity (mean *h* = 0.949) and low to moderate levels of nucleotide diversity (0.8% to 2.6%) based on the mtDNA control region fragment screened. Similar patterns have also been reported in other conspecifics including *A. Schlegelii* (*h* = 0.800 to 0.980; $\pi$ = 0.69 to 1.32%) [2] and *A. latus* (*h* = 0.951 to 0.995; $\pi$ = 2.3 to 2.6%) [53]. This general pattern may result from rapid population expansion events over historical time intervals, where populations do not also accumulate large nucleotide variation as a consequence of low effective population sizes [54].

### 4.2. Population Demography and Pleistocene Glaciations Impacts

All sampled populations also showed non-significant results for Tajima's *D* implying that accumulation of neutral variation and selection had not impacted them, the implication being that they had likely experienced population contractions from previously much higher levels. We also recorded however, a significant negative Fu's *F*s estimate for the Philippines sample resulting from an overabundance of low frequency (rare) haplotypes suggesting a possible signature of recent demographic expansion following a bottleneck. Both Funaura (AFP) and Nakamagawa (INP) populations exhibited unimodal distributions in the mismatch distribution analysis suggesting recent population expansions following a bottleneck [50]. All sampled populations in the present study appear to have experienced population expansions at some stage as indicated by the lack of significant SSD values and their individual raggedness indices [55]. In contrast, the Philippine population showed a bimodal distribution in the mismatch distribution analysis indicating that the population had been stable over recent times. This difference in population demographic patterns between the two regions (Iriomotejima Island vs the Philippines) may result potentially from a shorter spawning period (late winter season) in the Iriomotejima populations compared with that of the Philippines population. Higher mean ocean temperatures may increase the opportunity to recruit more individuals in the Philippines. Furthermore, the Iriomotejima populations also showed no significant variations in effective population size overtime based on the skyline plot analysis indicatingtheir limited dispersal abilities. This might be the result of small population size combined with a less favourable local environment that promotes a non-expanding demographic signal [56].

Two haplotype sub-networks were apparent in the minimum spanning haplotype network (MSN) analysis indicating potentially two sympatric lineages of *A. pacificas* across the sampled region. The NJ phylogeographic tree also suggested two lineages across the sampled range. Estimated divergence times for the two lineages from a common ancestor were traced back to the Pleistocene period (30,000–35,000 ybp). Significant sea level fluctuations and major climatic change occurred during the Pleistocene, in particular impacts were significant across the northern western Pacific promoting population divergence of many marine/intertidal species [57,58]. During the Quaternary (i.e., 2.4 million years before (m ybp) to 10,000 ybp (k ybp) several major ice age events reduced sea levels across this region and had major impacts on marine species distributions [59]. Paleo-climatic oscillations remarkably changed the mean seawater temperatures and changes to exposed continental shelves across this region impacted both ocean currents and land connections during the last glacial maxima (LGM). During this period, sea levels were low and up to 120–140 m lower than the present resulting in a sharp decline in coral reef distributions concomitant with drastic reduction and displacement of coral fish populations due to loss of habitats [59,60]. Different environmental conditions that included stabilization of mean seawater temperature and sea level occurred at the end of the LGM period. As a consequence, at this time many marine species are believed to have re-expanded into previously unavailable habitat (land) or re-colonized areas of shallow sea after glacial impacts were reduced. Some coastal waters that remained over the glacial periods provided refugia for marine species that apparently allowed them to persist until the late Pleistocene period when conditions warmed [61]. Modern Iriomotejima *A. pacificus* populations may have originated via colonization from the Philippines at the time mean sea water temperatures returned to pre-glacial levels. Potentially, this stock could then have recolonized pre-Pleistocene areas on the marginal sea (i.e., new islands) as glaciers receded. Similar scenarios have been offered for historical phylogeographic patterns in a number of co-distributed coastal marine fish taxa across the west Pacific region including: Spotted Sea bass, *Lateolabrax maculates* [62], Japanese Spanish mackerel, *Scomberomorus niphonius* [63] and Little Spinefoot, *Siganus spinus* [8]. Despite marine conditions improving for subtropical/tropical marine taxa after the LGM, rapid extensive expansions of *A. pacificus* populations have apparently not occurred across the sampled region. This could be the result of larvae having evolved traits that favour local retention. This potentially restricted *A. pacificus* populations from

demographic expansion. The Philippine *A. pacificus* populations may have experienced a rapid population expansion as Luzon Island is located in the southernmost region where the target species is now restricted to in SE Area. This region is far from the Iriomotejima island, the major focus of the present study. While in general, it can be a very complex process relating any particular paleo-climatic events to interpret the demographic history of any species [62]. Developing knowledge about a species' population genetic characteristics can assist in many ways. In particular, the demographic histories of marine fishes along the Kuroshio current the west Pacific is obviously very complex but deciphering them will be indispensable to formulate better hypotheses about the history of formation of marine fish fauna across this region.

*4.3. Population Genetic Structure*

In general, it is thought that most marine organisms will show little or no significant genetic differentiation over large geographical scales because most have pelagic larvae of long duration during their early life cycle which, in theory, will promote long distance dispersal with ocean currents, connecting dispersed populations. Significant genetic sub-division was evident however, among populations of *A. pacificus* between Iriomotejima Island in Japan and the Philippines in the present study.

Population structure can develop in many species due to interactions between physical and biological processes in the marine environment. Despite potential for long dispersal, a variety of physical mechanisms includingocean current and fronts can greatly influence genetic differentiation of populations in the sea [5]. Ocean currents in particular are considered to be a key factor in driving structured populations of coastal species in the NW Pacific region [64]. Genetic differentiation between Philippine and Iriomotejima populations of *A. pacificus* potentially result from impacts of the Kuroshio current which may act as a barrier to population connectivity, leading to genetic differentiation as it may be difficult for many marine organisms including fish species to pass through this strong current across this region [20–24]. The Kuroshio current is a strong boundary current that starts off in the Philippine islands and flows between Taiwan and the Ryukyu Archipelago in Japan moving north-eastward, essentially isolating the Ryukyu archipelago from the Philippine Islands. Modern distributions and patterns of genetic subdivision of Okinawa seabream (*A. sivicolus*) and black sea bream (*A. schlegelii*) on the main island of Japan have also been explained by this biogeographical isolation where both species are believed to have originated from a recent common ancestor [42].

It is also worthy to note that our pairwise $F_{ST}$ results also indicated some genetic differentiation between neighbouring populations within Iriomotejima Island (between Funaura bay and Nakamagawa coastal river populations). Genetic differentiation between intra-island populations has also been reported for other coastal organisms across the Ryukyu Archipelago including snail [65] and crab [66] species.

One possible reason for divergence between local populations within Iriomotejima Island may result from a high local larval self-recruitment rate, a factor that is known to play a significant role in other marine taxa of conservation status [67,68]. In some coastal species with long-lived marine pelagic larvae, studies have shown that a large proportion of larvae never disperse from, and remain close to, their parental population following adult spawning. Local self-recruitment may not only be promoted by physical retention factors but also facilitated by local adaptive traits of larvae because geographical distance between remote populations may not facilitate colonization and settlement in new environments due to habitat selection [69]. High self-recruitment rates have been reported in a number of marine fish taxa with larvae that reside inshore, have short planktonic larval duration and/or low adult mobility. These include a number of species in the family Gobiesociadae, Syngnathidae [70] and Tripterygiidae [71], suggesting that the extent of effective dispersal between regions can be much lower than is currently often assumed. Moreover, spawning is also often influenced by the intensity of local winds and currents. So, if larvae hatch at a

time of low current speed, this can significantlyreduce transportation of larvae over long distances promoting local settlement and enhancing population self-recruitment [13].

## 5. Conclusions

Results of the present study of *A. pacificus* population structure between populations of conservation status on Iriomotejima Island in Japan and the Philippines showed strong evidence for both significant regional differentiation but also between local populations on the same island in the West Pacific region. High levels of genetic diversity were observed in all sampled populations, suggesting that large effective population sizes may have been maintained at least in modern times. The degree of genetic differentiation between local populations within Iriomotejima Island reflecting limited local gene flow may result from local larval self-recruitment and/or on-shelf current circulation patterns. Moreover, two distinct lineages were identified amongst the *A. pacificus* populations sampled here. We speculated that climatic fluctuations during the Quaternary glacial period may have had an important impact on the *A. pacificus* phylogeographic patterns reported here. This knowledge may assist in future designation of potential population management units and in formulating better fisheries management and conservation plans for this important regional species. To develop a deeper understanding of the level of population connectivity and dispersal mechanisms in *A. pacificus*, the use of highly polymorphic nuclear genome loci (e.g., microsatellites) is recommended in future studies.

**Supplementary Materials:** The following supporting information can be downloaded at: https://www.mdpi.com/article/10.3390/d14121117/s1, Table S1: Haplotype composition of all three localities of *A. pacificus*.

**Author Contributions:** Conceptualization: M.R.-U.I., K.T. and H.I.; methodology: M.R.-U.I. and H.I.; data curation: M.R.-U.I. and H.I.; formal analysis and writing—original draft: M.R.-U.I.; writing—review and editing: M.R.-U.I., K.T. and H.I.; resources: H.I.; project administration and funding acquisition: M.R.-U.I. and H.I.; Supervision: H.I. and K.T. All authors have read and agreed to the published version of the manuscript.

**Funding:** This research was funded by the grant for JSPS postdoctoral fellowship, grant number P19094.

**Institutional Review Board Statement:** Not applicable.

**Informed Consent Statement:** Not applicable.

**Data Availability Statement:** Data is publicly accessible at GenBank under accession number LC643484-LC643555 (https://www.ncbi.nlm.nih.gov/nuccore/ (accessed on 3 July 2022).

**Acknowledgments:** We would like to deeply thank Takuto Sumi, Ifue Fukuchi, Hiroto Nagai for help with sample collection. Thanks to Bernei Gracines for arrangement of the sampling trip in the Philippines. Thanks to Eko Hardianto for assistance with genetic laboratory work. Thanks to Peter Mather for English check. We would like to thank Yoshihiro Kanamori and Yoshigoro Imoto for their assistance with the gill net survey by boat. We would like to acknowledge the Japan Society for Promotional Science (JSPS) for providing research funds through a postdoctoral fellowship award (Md Rakeb-Ul Islam).

**Conflicts of Interest:** The authors declare no conflict of interest.

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
