# Peer review of "Cryptic Lineage and Genetic Structure of Acanthopagrus pacificus Populations in a Natural World Heritage Site Revealed by Population Genetic Analysis"

_diversity, doi:10.3390/d14121117_

Round 1

Reviewer 1 Report

The article presents interesting results which are of importance for future management of the studied fish species as well as other species.

As a referee, I must say that I do not master the laboratory methods used, nor am I used to the analysis program mega.

Here are some details to the manuscript:

Line 90: were they adult speciemens?

Line 92: two different type sizes in the text?

Line 177: “populations pairs” should be “population pairs”

Line 182: three decimals will do: FST = 0.155

Line 210 – 214: rewrite to shorten down period length

Line 215: comments as above

Line 233: “120 - 140m lower» -> 120 – 140 m lower

Line 239: change «impacts had reduced» -> “glacial impacts were reduced”

Line 313: “at least modern times» -> “at least in modern times”

Author Response

Thank you very much for your insightful comments. Here I addressed your comment in word file below as attached file.

Reviewer 2 Report

Islam et al. investigated the population genetic structure of Acanthopagrus pacificus using genetic data from the mtDNA control region in 159 individuals. Based on their results, the authors concluded a significant genetic subdivision and two distinct cryptic lineages in the species.

I have a few major concerns about the manuscript. First, I am not sure how they identified the species of their samples. If I am not mistaken, A. pacificus was first described by Iwatsuki et al. (2010, Bull. Natl. Mus. Nat. Sci., Ser. A, 36:115-130). And the authors of this paper mentioned that A. pacificus is most similar to A. berda. Is there any chance that one of the “two distinct cryptic lineages” is actually from A. berda? If not, what scientific evidence will support it? Related to this issue, I am puzzled by the statement in M&M “Samples were collected from Irimote (Iriomote?) island” whereas a part of their samples is actually from the Philippines.

              Another concern of mine is about the use of mtDNA for population genetic characteristics such as population expansion. Because of the unique characteristics of mtDNA (maternal inheritance, no recombination, etc.), there is a significant amount of discussion over the use of mtDNA as a population genetic marker (e.g., Ballard & Kreitman 1995 TREE 10:485-488; Ballard & Whitlock 2004 Mol. Ecol. 13:729-744; Hurst & Jiggins 2005 Proc. R. Soc. B 272: 1525-1534). I do not think that mtDNA is always inappropriate as a population genetic marker, but it should be used very carefully reflecting the discussions above. For example, how is it reasonable to assume neutrality of the mtDNA control region in fishes? The authors applied their data to some neutrality tests in the manuscript, but they used these tests to argue for population expansions rather than selective sweep on mtDNA. It will affect an argument that (historical) Ne of mitochondria is/is not consistent with that of the population estimated using nuclear gene. To support their arguments, therefore, evidence from additional genetic analyses, hopefully including both mtDNA and ncDNA markers, with more populations, should be provided.

Specific comments:

Line 62. “genetically homogenous or taxa with non-significant genetic structure” What is the nominative in this sentence?

Line 92. Font size difference Why?

Line 114. The current version of MEGA is v11 (Tamura et al. 2021). Is there any specific reason for using the MEGA v7 (six years old), and is there any difference in results between them?

Fig. 5. I do not understand where each sample was from. Besides, I see A. schiegeii was used as an outgroup. I do not know if this is the most closely related species or not, and I found no description of this choice in the main text.

Table 1. “k, mean pairwise” what is it?

Author Response

Thank you very much for your comments. Here I addressed your comment below as attached file.

Reviewer 3 Report

I consider the manuscript valuable, well and comprehensively prepared. Both the methods and the results are presented legibly and correctly. Very interesting conclusions.

Author Response

We thank the reviewer very much for his insightful comments. Here I addressed the comment below as attached file.

Reviewer 4 Report

The manuscript contains detailed research on the genetic diversity and structure of the Acanthopargus pacificus populations near to Iriomotejima Island and Luzon Island. It presents an interesting experiment and a well-built design. I have only one concern that needs to be answered and some minor suggestions before this manuscript can be accepted:

Results:

Line 382-385. Please check the result of the mismatch analysis as the opposite seems correct to me. Based on Figure 2, the pairwise mismatch distribution in the case of the Philippines seems to be unimodal rather than bimodal, while the other two populations are multimodal (bimodal). This is confirmed by the HRI index in Table 2. The Philippines had the lowest value. HRI index takes larger values for multimodal distributions commonly found in a stationary population than for unimodal and smoother distributions typical of expanding populations. This interpretation also corresponds to the value of Fu's index, which was significant in the case of the Philippines, and the Bayesian skyline plot analysis which shows a rapid increase in the Ne of the Philippines.

Lines 469-490 in the Discussion chapter should also be revised if this interpretation is acceptable.

minor suggestions:

Line 292 - Hsu et al 2011 should be numbered in the text and included in the references

Line 397 - please add in the methods, how the genetic divergence of the two sub-networks was calculated

please add an explanation for the Pooled part of Figure 3.

Author Response

Reviewer 4:

The manuscript contains detailed research on the genetic diversity and structure of the Acanthopargus pacificus populations near to Iriomotejima Island and Luzon Island. It presents an interesting experiment and a well-built design. I have only one concern that needs to be answered and some minor suggestions before this manuscript can be accepted.

Reply: Thank you very much for your insightful comments.

Point 1: Line 382-385 - Please check the result of the mismatch analysis as the opposite seems correct to me. Based on Figure 2, the pairwise mismatch distribution in the case of the Philippines seems to be unimodal rather than bimodal, while the other two populations are multimodal (bimodal). This is confirmed by the HRI index in Table 2. The Philippines had the lowest value. HRI index takes larger values for multimodal distributions commonly found in a stationary population than for unimodal and smoother distributions typical of expanding populations. This interpretation also corresponds to the value of Fu's index, which was significant in the case of the Philippines, and the Bayesian skyline plot analysis which shows a rapid increase in the Ne of the Philippines.

Point 2: Lines 469-490 - in the Discussion chapter should also be revised if this interpretation is acceptable.

Reply: Thank you very much for your nice comments. The HRI index value is not significant for all populations that is why we did not consider this HRI index value for interpretation of the mismatch analysis result. A line was mentioned in the discussion section -“A significant negative Fu’s Fs estimate for the Philippines sample resulting from an overabundance of low frequency (rare) haplotypes suggesting a possible signature of recent demographic expansion following a bottleneck.”

Point 3: Line 292 - Hsu et al 2011 should be numbered in the text and included in the references.

Reply: Done

Point 4: Line 397 - please add in the methods, how the genetic divergence of the two sub-networks was calculated

Reply: Thanks once again for your comment. It was mentioned in materials and methods section. The sentence was re-written “Genetic divergence between lineages were also calculated using Kimura’s two – parameter (K2P) model [41] in MEGA version 7 program [35]”.

Point 5: Please, add an explanation for the Pooled part of Figure 3.

Reply: Pooled samples means all the samples of three populations were pooled to run in BEAST v. 1.7.4 to see the changes in effective population size over time for the entire samples. It was mentioned in materials and methods section. “Bayesian skyline analyses, implemented in BEAST v. 1.7.4 [46] were then performed to estimate changes in effective population size over time for the entire or pooled sample.” Another sentence was rewritten in results section-“However, past demographic expansion of A. pacificus populations was observed in pooled sample for all sampled populations based on Bayesian skyline analyses.”